Original research

# Relationships between multiple patient safety outcomes and healthcare and hospital-related risk factors in colorectal resection cases: cross-sectional evidence from a nationwide sample of 232 German hospitals

Felix Walther [ID],[1,2] Jochen Schmitt,[2] Maria Eberlein-Gonska,[1] Ralf Kuhlen,[3] Peter Scriba,[3] Olaf Schoffer [ID],[2] Martin Roessler [ID],[2]

¹Quality and Medical Risk Management, University Hospital Carl Gustav Carus, Dresden, Germany
²Center for Evidence-based Healthcare, TU Dresden Faculty of Medicine Carl Gustav Carus, Dresden, Germany
³Initiative Qualitätsmedizin e.V, Berlin, Germany

**Correspondence to**
Felix Walther;
felix.walther@ukdd.de

## ABSTRACT

**Objectives** Studies analysing colorectal resections usually focus on a specific outcome (eg, mortality) and/or specific risk factors at the individual (eg, comorbidities) or hospital (eg, volume) level. Comprehensive evidence across different patient safety outcomes, risk factors and patient groups is still scarce. Therefore the aim of this analysis was to investigate consistent relationships between multiple patient safety outcomes, healthcare and hospital risk factors in colorectal resection cases.

**Design** Cross-sectional study.

**Setting** German inpatient routine care data of colorectal resections between 2016 and 2018.

**Participants** We analysed 54 168 colon resection and 20 395 rectum resection cases treated in German hospitals. The German Inpatient Quality Indicators were used to define colon resections and rectum resections transparently.

**Primary outcome measures** Additionally to in-hospital death, postoperative respiratory failure, renal failure and postoperative wound infections we included multiple patient safety outcomes as primary outcomes/dependent variables for our analysis. Healthcare (eg, weekend surgery), hospital (eg, volume) and case (eg, age) characteristics served as independent covariates in a multilevel logistic regression model. The estimated regression coefficients were transferred into ORs.

**Results** Weekend surgery, emergency admissions and transfers from other hospitals were significantly associated (ORs ranged from 1.1 to 2.6) with poor patient safety outcome (ie, death, renal failure, postoperative respiratory failure) in colon resections and rectum resections. Hospital characteristics showed heterogeneous effects. In colon resections hospital volume was associated with insignificant or adverse associations (postoperative wound infections: OR 1.168 (95% CI 1.030 to 1.325)) to multiple patient safety outcomes. In rectum resections hospital volume was protectively associated with death, renal failure and postoperative respiratory failure (ORs ranged from 0.7 to 0.8).

### STRENGTHS AND LIMITATIONS OF THIS STUDY

⇒ Large and current sample providing a broad span of cases, hospital types, ownerships and locations.
⇒ Comprehensive analysis of multiple patient safety outcomes and multiple sets (case, healthcare, hospital) of risk factors.
⇒ Use of previously validated outcomes that were reported to occur most likely during hospitalisation.
⇒ Accounting data lack information on patient history, medication, length of anaesthesia, staff-to-patient ratios, surgeon volumes, centralisation and which of the coded diagnoses had been present on admission.

**Conclusions** Transfer from other hospital and emergency admission are constantly associated with poor patient safety outcome. Hospital variables like volume, ownership or localisation did not show consistent relationships to patient safety outcomes.

**Trial registration number** ISRCTN10188560.

## INTRODUCTION

Measuring, assuring and improving patient safety are important objectives regarding patient outcome, payment and accreditation in colorectal resections. One of the most frequently used outcome indicator in colorectal resections is in-hospital mortality.[1] However, it has been stressed that patient safety is reflected in both mortality and non-mortality outcomes.[2,3] Therefore the measurement of outcomes beyond mortality is necessary for a comprehensive assessment of patient safety and care quality.[1] Additionally various risk factors for a poor patient safety outcome were analysed in previous studies. Besides patient characteristics (eg, age, sex, comorbidities), especially the influence of

healthcare (eg, weekend surgery, emergency, transfer from other hospital) and hospital variables (eg, volume, urbanisation degree) were widely discussed. Weekend surgery,[4–6] emergency admission,[7 8] transfer from other hospitals[9–11] and case volume[12 13] were found to have significant effects on mortality in colorectal resections. Analogous to patient outcomes, previous studies usually considered only subsets of these risk factors without analysing them together.

A comprehensive analysis of patient safety and its covariates in colorectal resections should take multiple outcomes and multiple risk factors into account.[14] To our knowledge, such comprehensive analyses have rarely been reported. Based on that assumption, our analysis aimed to investigate whether healthcare and hospital characteristics are associated with multiple patient safety outcomes in colorectal resections. Using a 3-year sample (2016–2018) of German inpatient claims data we investigated relationships between case, healthcare and hospital characteristics and the patient safety outcomes in-hospital death, postoperative respiratory failure, renal failure and post-operative wound infections in colorectal resections.

## MATERIALS AND METHODS
This explorative cross-sectional analysis was embedded into the IMPRESS study. The IMPRESS study was a cluster-randomised trial evaluating the effects of clinical peer review on mortality in patients ventilated >24 hours nested in a prospective cohort study of 232 participating hospitals. Details, baseline, explorative and confirmatory results of the IMPRESS study were published previously.[15–18] The study has been registered at ISCRTN.[19] The identification of possible covariates of mortality and non-mortality outcomes in colorectal resections was a secondary aim of the IMPRESS study.

### Data sources
The data used in this study were derived from two routine data sets. We used claims data according to German law regulating inpatient claims data (§21 Krankenhausentgeltgesetz) to gather information concerning age, sex, reason of admission, discharge destination, diagnoses/comorbidities (International Classification of Diseases, 10th Revision - German Modification (ICD-10-GM)) and medical/surgical procedures (Operationen- und Prozedurenschlüssel (OPS) codes). We applied the predefined groups of the Elixhauser comorbidity index and its coding modifications for ICD-10 (online supplemental file S1) to adjust for relevant comorbidities. The Elixhauser comorbidity index is a score used to adjust for chronic or non-acute comorbidities in routine data sets.[20 21] To assess hospital characteristics (ownership, university hospital status, urbanisation) we used data from the German hospital register ('Deutsches Krankenhausverzeichnis').

### Study participation and privacy
All participating hospitals submitted a written consent regarding participation prior to the start of the IMPRESS study. The data trust site at Koordinierungszentrum für

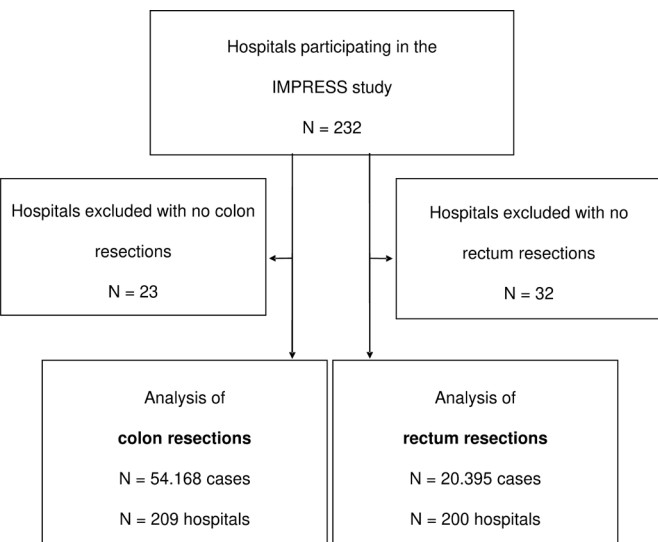

**Figure 1** Flowchart of hospitals included for analysis.

Klinische Studien (KKS) Dresden ensured the anonymisation of the data. The Center for Evidence-Based Healthcare (ZEGV) Dresden analysed the anonymised data.

### Patient and public involvement
This cross-sectional analysis used observational routine data based on predefined outcomes and covariates without intervention and did not involve patients or the public in the design, conduct, reporting or dissemination plans of our research.

### Population
Overall, 232 hospitals participated in the IMPRESS study (figure 1). We included all cases with a colon and/or rectum resection in the participating hospitals in 2016–2018. Due to anonymisation, the data do not contain a patient-relation. Therefore patients admitted more than once entered the analysis as multiple hospital cases. For each hospital case, all of the documented information in terms of diagnoses and medical/surgical procedures during hospitalisation was available. We used the definitions of the German Inpatient Quality Indicators to define and distinguish partial colon resections (online supplemental file S2), total colon resections (online supplemental file S3) and rectum resections (online supplemental file S4).[22] Hospitals without colon or rectum resections were excluded.

### Outcomes and covariates
Following evidence from a previous study, in-hospital death, respiratory failure, renal failure and wound infection can be validly operationalised in hospital discharge data.[23] Hence, we analysed these outcomes in accordance with previously tested case definitions as presented in online supplemental file S5. The outcomes death, postoperative respiratory failure, renal failure and postoperative wound infections were selected as dependent variables.

The independent variables were classified into three groups:
1. Case (age, sex, Elixhauser comorbidities).[20 21]

2. Healthcare (admission date, surgical procedures/OPS codes, reason for admission, discharge destination).
3. Hospital (case volume, ownership, university hospital status, urbanisation degree rural/urban).

This study focused on healthcare and hospital variables. Case variables were primarily used for adjustment.

To adjust for all potentially relevant risk factors available in the data, the estimations included the full set of independent case, healthcare and hospital variables. Case level included age, sex and Elixhauser comorbidities.[21] Healthcare level included admission reason (referral/emergency case/transfer from other hospital), weekend surgery (identifiable via time stamp of the procedure) and total resection of the colon and resections of the colon and rectum. Hospital level included hospital case volume, degree of urbanisation (rural/urban), university hospital status and ownership (public/non-profit/private). Case volume entered the regression models in logarithmic form. This transformation of hospital volume data captures that volume-outcome relationships may be more pronounced at low case volumes.[24]

## Statistical methods
We described case, healthcare and hospital characteristics using absolute and relative frequencies in case of categorical variables. For continuous variables, we reported median, first and third quartile. Relationships between patient safety outcomes and case-level, healthcare-level and hospital-level risk factors were estimated using multilevel logistic regression models. These models contained a random intercept at the hospital level to capture the correlation of patient outcomes within hospitals.[25] Estimations were conducted separately for cases with colon and rectum resection in bivariate and multivariate analyses. To improve interpretability of estimated effect sizes, we transformed the estimated regression coefficients into ORs. An estimated effect was considered statistically significant if its p value was below 5%. Statistical analysis was performed using Stata V.15.1.

## Sensitivity analysis
In the preliminary research it became apparent that the literature distinguishes between colorectal cancer surgery and general colorectal resections.[1 12 13 26 27] Therefore, with respect to possible effect modifications, we explored differences between cases with and without a cancer diagnosis. The same applies to reported interactions between admission reason and the status of university hospitals compared with non-university hospitals.[16] Therefore we also reviewed these interactions to detect possible effect modifications.

## RESULTS
Overall, 71 060 cases with colon and/or rectum resection were included in the analysis. Separating colon resections and rectum resections a total of 54 168 colon resection cases were treated in 209 hospitals. In total, 20 395 cases of rectum resections were treated in 200 hospitals. If both colon and rectum resection were documented (3503 cases), the case was analysed for both groups.

The minority of included cases received combined colorectal resections (partial and total colon, colon and rectum). Emergency case admission or transfer from other hospitals were less frequent than referral. Compared with rectum resections, colon resections were more often surgically treated on weekends (8.6% vs 3.8%), admitted as an emergency case (29.7% vs 18.3%) or transferred from other hospitals (3.5% vs 1.9%). The same applies to the rate of poor patient safety outcomes. Colon resection cases revealed higher rates of in-hospital death (9.6% vs 4.2%), postoperative respiratory failure (16.7% vs 12.2%), renal failure (15.2% vs 10.3%) and post-operative wound infections (11.3% vs 11.2%) than rectum resection cases.

The majority of the analysed hospitals were localised in urban regions (59%). Most were in private (40%) or public (40%) ownership. The annual median hospital case volume was 72 (Q1=38; Q3=116) for colon and 26 (Q1=11; Q3=42) for rectum resections.

The median age ranged from 67 to 68 years (table 1). Male and female sex in colon and rectum resections were approximately equally represented. For Elixhauser comorbidities, the most frequent codes were solid tumour without metastasis (colon: 47.3%, rectum: 67.0%), uncomplicated hypertension (colon and rectum: 47.6%) and fluid and electrolyte disorders (colon: 45.2%, rectum: 40.5%) in both procedure groups. Descriptive results for all Elixhauser comorbidities are presented in online supplemental file S6.

The bivariate analysis provided in online supplemental files S7 and S8 was performed to identify unadjusted effects of single covariates on the outcomes. The following multivariable analysis, focusing on healthcare and hospital level (tables 2 and 3), was performed to achieve adjusted and robust effects.

## Healthcare covariates
Admission as an emergency case or transfer from another hospital were associated to multiple poor patient safety outcomes in both groups. For example, higher odds of in-hospital death were related to emergency admission in colon (OR 1.84 (95% CI 1.69 to 2.01) and rectum resections (OR 2.02 (95% CI 1.67 to 2.45) compared with referral hospital admissions. The same applies to transfer from other hospital, the odds of in-hospital death were higher in colon (OR 2.52 (95% CI 2.19 to 2.91)) and rectum resections (OR 2.67 (95% CI 1.87 to 3.82)). Except of postoperative wound infections, weekend surgery was associated with worsened patient safety outcome in both groups.

## Hospital covariates
While most of the healthcare-level covariates showed similar associations in both groups, hospital covariates showed insignificant or heterogeneous effects.

A higher annual case volume of colon resections indicated a higher risk of postoperative wound infections (OR 1.16 (95% CI 1.03 to 1.32)). The remaining associations between annual case volume of colon resections

**Table 1** Case and hospital characteristics of colon and rectum resections

| | Colon resection | | Rectum resection | |
|---|---|---|---|---|
| | n | % / Q1; Q3 | n | % / Q1; Q3 |
| Number of cases | 54 168 | (100.0) | 20 395 | (100.0) |
| **Patient safety outcomes** | | | | |
| In-hospital death | | | | |
| No | 48 914 | (90.31) | 19 525 | (95.73) |
| Yes | 5254 | (9.68) | 870 | (4.26) |
| Postoperative respiratory failure | | | | |
| No | 45 074 | (83.21) | 17 901 | (87.77) |
| Yes | 9094 | (16.78) | 2494 | (12.22) |
| Renal failure | | | | |
| No | 45 920 | (84.77) | 18 279 | (89.62) |
| Yes | 8248 | (15.22) | 2116 | (10.37) |
| Postoperative wound infections | | | | |
| No | 48 013 | (88.63) | 18 109 | (88.79) |
| Yes | 6155 | (11.36) | 2286 | (11.20) |
| **Healthcare characteristics** | | | | |
| Colon resection | | | | |
| Total | 2662 | (4.91) | – | – |
| Partial | 51 310 | (94.72) | – | – |
| Both | 196 | (0.36) | – | – |
| Rectum resection | | | | |
| No | 50 665 | (93.53) | – | – |
| Yes | 3503 | (6.46) | 20 395 | (100.00) |
| Colon and rectum resection | | | | |
| No | 50 665 | (93.53) | 16 892 | (82.82) |
| Yes | 3503 | (6.46) | 3503 | (17.17) |
| Weekend surgery | | | | |
| No | 49 473 | (91.33) | 19 603 | (96.11) |
| Yes | 4695 | (8.66) | 792 | (3.88) |
| Admission reason | | | | |
| Referral | 36 129 | (66.69) | 16 249 | (79.67) |
| Emergency case | 16 116 | (29.75) | 3744 | (18.35) |
| Transfer from other hospital | 1923 | (3.55) | 402 | (1.97) |
| **Hospital characteristics** | | | | |
| Hospitals included | 209 | (100.00) | 200 | (100.00) |
| Annual volume | | | | |
| Colon resection cases (median) | 72 | (38; 119) | – | – |
| Total colon resection (median) | 1 | (0; 3) | – | – |
| Rectum resections (median) | – | – | 26 | (11; 42) |

Continued

**Table 1** Continued

| | Colon resection | | Rectum resection | |
|---|---|---|---|---|
| | n | % / Q1; Q3 | n | % / Q1; Q3 |
| Urbanisation | | | | |
| Urban | 124 | (59.33) | 119 | (59.50) |
| Tural | 85 | (40.66) | 81 | (40.50) |
| Ownership | | | | |
| Public | 82 | (39.23) | 80 | (40.00) |
| Non-profit | 41 | (19.61) | 39 | (19.50) |
| Private | 86 | (41.14) | 81 | (40.50) |
| University hospital | | | | |
| No | 201 | (96.17) | 192 | (96.00) |
| Yes | 8 | (3.82) | 8 | (4.00) |
| **Case characteristics** | | | | |
| Age | | | | |
| Median | 68 | (56; 77) | 67 | (57; 77) |
| Sex | | | | |
| Male | 26 954 | (49.76) | 10 367 | (50.83) |
| Female | 27 214 | (50.23) | 10 028 | (49.16) |
| Elixhauser comorbidities (…)* | | | | |

Q1: first quartile. Q3: third quartile.
*Results of Elixhauser comorbidities (eg, alcohol abuse, blood loss anaemia, cardiac arrhythmias…) are presented in online supplemental file S6.

and patient safety outcomes were insignificant. A higher annual volume of rectum resections was associated with lower risks of in-hospital death (OR 0.70 (95% CI 0.61 to 0.80)), postoperative respiratory failure (OR 0.84 (95% CI 0.72 to 0.98)) and renal failure (OR 0.85 (95% CI 0.76 to 0.95)).

Rural localisation showed lower odds of renal failure (OR 0.77 (95% CI 0.63 to 0.93)) in cases with only colon resections.

Treatment in university hospitals was associated with increased odds of postoperative wound infections in colon (OR 1.98 (95% CI 1.17 to 3.35)) and rectum resections (OR 2.29 (95% CI 1.35 to 3.86)) compared with treatment in non-university hospitals.

The hospital ownership revealed differences between both groups and patient safety outcomes. Non-profit (OR 0.74 (95% CI 0.55 to 0.99)) or private (OR 0.77 (95% CI 0.60 to 0.99)) ownership was associated with lower risks of postoperative wound infections in colon resections. In contrast, odds of in-hospital death (OR 1.24 (95% CI 1.02 to 1.50)) and renal failure (OR 1.93 (95% CI 1.56 to 2.40)) in colon resections were higher in private hospitals. Rectum resections did not show significant associations of ownership and patient safety outcomes except for higher odds of renal failure (OR 1.59 (95% CI 1.25 to 2.03)) in private hospitals.

**Table 2** Multivariate analysis of patient safety outcomes in 54 168 colon resections in 209 hospitals

| | In-hospital death | | Postoperative respiratory failure | | Renal failure | | Postoperative wound infection | |
|---|---|---|---|---|---|---|---|---|
| | OR | 95% CI | OR | 95% CI | OR | 95% CI | OR | 95% CI |
| **Healthcare covariates** | | | | | | | | |
| Admission reason | | | | | | | | |
| Referral | Ref. | | Ref. | | Ref. | | Ref. | |
| Emergency case | 1.847*** | (1.692 to 2.015) | 1.413*** | (1.320 to 1.513) | 1.453*** | (1.349 to 1.566) | 1.145*** | (1.067 to 1.228) |
| Transfer from other hospital | 2.528*** | (2.193 to 2.915) | 1.982*** | (1.749 to 2.245) | 1.908*** | (1.678 to 2.171) | 1.223** | (1.071 to 1.397) |
| Weekend surgery | | | | | | | | |
| No | Ref. | | Ref. | | Ref. | | Ref. | |
| Yes | 1.669*** | (1.515 to 1.839) | 1.426*** | (1.312 to 1.550) | 1.480*** | (1.360 to 1.610) | 1.080 | (0.984 to 1.186) |
| Total colon resection | | | | | | | | |
| No | Ref. | | Ref. | | Ref. | | Ref. | |
| Yes | 2.679*** | (2.369 to 3.029) | 1.639*** | (1.472 to 1.825) | 2.228*** | (1.999 to 2.483) | 1.022 | (0.913 to 1.143) |
| Colon and rectum resection | | | | | | | | |
| No | Ref. | | Ref. | | Ref. | | Ref. | |
| Yes | 1.103 | (0.960 to 1.267) | 1.524*** | (1.378 to 1.686) | 1.408*** | (1.265 to 1.567) | 1.579 | (1.426 to 1.748) |
| **Hospital covariates** | | | | | | | | |
| Case volume | 0.968 | (0.871 to 1.076) | 0.919 | (0.807 to 1.047) | 0.992 | (0.891 to 1.106) | 1.168* | (1.030 to 1.325) |
| Area | | | | | | | | |
| Urban | Ref. | | Ref. | | Ref. | | Ref. | |
| Rural | 1.061 | (0.893 to 1.261) | 0.863 | (0.648 to 1.149) | 0.772** | (0.635 to 0.939) | 1.032 | (0.824 to 1.292) |
| University hospital | | | | | | | | |
| No | Ref. | | Ref. | | Ref. | | Ref. | |
| Yes | 1.303 | (0.888 to 1.912) | 0.687 | (0.338 to 1.397) | 1.412 | (0.889 to 2.241) | 1.981* | (1.171 to 3.352) |
| Ownership | | | | | | | | |
| Public | Ref. | | Ref. | | Ref. | | Ref. | |
| Non-profit | 1.012 | (0.811 to 1.262) | 1.057 | (0.726 to 1.540) | 0.867 | (0.670 to 1.122) | 0.744* | (0.555 to 0.998) |
| Private | 1.244* | (1.026 to 1.507) | 1.329 | (0.972 to 1.817) | 1.937*** | (1.563 to 2.400) | 0.777* | (0.608 to 0.992) |
| **Case covariates** | | | | | | | | |
| Sex | | | | | | | | |
| Male | Ref. | | Ref | | Ref | | Ref | |
| Female | 0.937 | (0.873 to 1.006) | 0.788*** | (0.745 to 0.833) | 0.683*** | (0.645 to 0.725) | 0.882*** | (0.832 to 0.936) |
| Age | 1.050*** | (1.046 to 1.053) | 1.014*** | (1.012 to 1.017) | 1.024*** | (1.021 to 1.026) | 0.998 | (0.996 to 1.000) |
| Elixhauser comorbidities (…)† | | | | | | | | |

***P<0.001, **P<0.01, *p<0.05.
†Results of Elixhauser comorbidities (eg, alcohol abuse, blood loss anaemia, cardiac arrhythmias, chronic pulmonary disease…) are presented in online supplemental file S9.

**Table 3** Multivariate analysis of patient safety outcomes in 20 395 rectum resections in 200 hospitals

| | In-hospital death | | Postoperative respiratory failure | | Renal failure | | Post-operative wound infection | |
|---|---|---|---|---|---|---|---|---|
| | OR | 95% CI | OR | 95% CI | OR | 95% CI | OR | 95% CI |
| **Healthcare covariates** | | | | | | | | |
| Admission reason | | | | | | | | |
| Referral | Ref. | | Ref. | | Ref. | | Ref. | |
| Emergency case | 2.028*** | (1.675 to 2.454) | 1.335*** | (1.170 to 1.523) | 1.342*** | (1.169 to 1.540) | 1.291*** | (1.138 to 1.466) |
| Transfer from other hospital | 2.679*** | (1.874 to 3.828) | 1.859*** | (1.406 to 2.459) | 1.927*** | (1.461 to 2.541) | 1.484** | (1.131 to 1.948) |
| Weekend surgery | | | | | | | | |
| No | Ref. | | Ref. | | Ref. | | Ref. | |
| Yes | 1.960*** | (1.483 to 2.591) | 1.427** | (1.150 to 1.770) | 1.391** | (1.127 to 1.717) | 0.985 | (0.784 to 1.238) |
| Total colon resection | | | | | | | | |
| No | Ref. | | Ref. | | Ref. | | Ref. | |
| Yes | 2.579*** | (2.163 to 3.074) | 2.164*** | (1.921 to 2.438) | 1.859*** | (1.645 to 2.100) | 1.522*** | (1.356 to 1.708) |
| **Hospital covariates** | | | | | | | | |
| Case volume | 0.703*** | (0.611 to 0.809) | 0.844* | (0.725 to 0.982) | 0.853** | (0.760 to 0.958) | 0.973 | (0.854 to 1.109) |
| Area | | | | | | | | |
| Urban | Ref. | | Ref. | | Ref. | | Ref. | |
| Rural | 1.072 | (0.817 to 1.407) | 0.904 | (0.639 to 1.281) | 0.834 | (0.670 to 1.037) | 0.854 | (0.663 to 1.101) |
| University hospital | | | | | | | | |
| No | Ref. | | Ref. | | Ref. | | Ref. | |
| Yes | 1.616 | (0.979 to 2.665) | 0.853 | (0.379 to 1.920) | 1.299 | (0.829 to 2.037) | 2.292** | (1.358 to 3.869) |
| Ownership | | | | | | | | |
| Public | Ref. | | Ref. | | Ref. | | Ref. | |
| Non-profit | 0.851 | (0.614 to 1.179) | 1.059 | (0.674 to 1.664) | 0.762 | (0.576 to 1.008) | 0.761 | (0.553 to 1.048) |
| Private | 0.925 | (0.682 to 1.254) | 1.267 | (0.865 to 1.858) | 1.597*** | (1.256 to 2.030) | 0.846 | (0.642 to 1.117) |
| **Case covariates** | | | | | | | | |
| Sex | | | | | | | | |
| Male | Ref. | | Ref | | Ref | | Ref | |
| Female | 0.842* | (0.710 to 0.998) | 0.842** | (0.759 to 0.934) | 0.685*** | (0.613 to 0.765) | 0.826*** | (0.748 to 0.912) |
| Age | 1.068*** | (1.059 to 1.078) | 1.014*** | (1.009 to 1.018) | 1.021*** | (1.016 to 1.026) | 0.998 | (0.995 to 1.002) |
| Elixhauser comorbidities (…)† | | | | | | | | |

***P<0.001, **p<0.01, *p<0.05.

†Results of Elixhauser comorbidities (eg, alcohol abuse, blood loss anaemia, cardiac arrhythmias, chronic pulmonary disease…) are presented in online supplemental file S10.

## Case covariates

Female sex was consistently associated with better outcomes in both groups, except of a borderline-insignificant association with in-hospital death (OR 0.93 (95% CI 0.87 to 1.00)) in colon resections. Age was associated with higher risks of in-hospital death, postoperative respiratory failure and renal failure in both groups. Regarding postoperative wound infections, age was a borderline-insignificant protective factor in colon (OR 0.99 (95% CI 0.99 to 1.00)) and rectum (OR 0.99 (95% CI 0.99 to 1.00)) resections. Of all Elixhauser comorbidities analysed, coagulopathies showed the highest ORs for poor patient safety outcomes including higher ORs of death in colon (OR 4.17 (95% CI 3.864 to 4.509)) or rectum (OR 4.30 (95% CI 3.600 to 5.158)) resections. The same applies to other patient safety outcomes like postoperative respiratory failure in colon (OR 3.117 (95% CI 2.920 to 3.327)) or rectum (OR 3.052 (95% CI 2.697 to 3.455)) resections, renal failure in colon (OR 3.332 (95% CI 3.118 to 3.561)) and rectum (OR 2.886 (95% CI 2.541 to 3.277)) resections and postoperative wound infections in colon (OR 1.644 (95% CI 1.531 to 1.764)) or rectum (OR 1.770 (95% CI 1.570 to 1.996)) resections. Along with coagulopathies, fluid and electrolyte disorders, peripheral vascular disorders, congestive heart failure, chronic pulmonary disease, cardiac arrhythmias and pulmonary circulation disorders were also associated with multiple poor patient outcomes in both procedure groups. The multivariate results for the remaining Elixhauser groups can be found in online supplemental files S9 and S10.

We reviewed differences in results of stratified analyses for cases with and without cancerous colon and rectum resections. Significant effect reversals were not observed (online supplemental files S11–S14). The review also did not reveal differences between university and non-university hospitals in terms of emergency admission or transfer from other hospital (online supplemental files S15 and S16). Therefore, a stratification between cases with and without cancer or university and non-university hospitals has not been applied.

## DISCUSSION

This large cross-sectional analysis of 54 168 colon resections and 20 395 rectum resections presents new and comprehensive findings for patient safety.

Healthcare-level covariates were significant risk factors for multiple patient safety outcomes. Preoperative transfer from other hospitals and emergency admission as possible proxy for case urgency were precursors of poor patient safety outcome in both groups. These findings confirm recent literature reporting associations between emergency admissions or transfers from other hospitals and 30-day-mortality, 5-year survival, complications, length of stay or morbidities.[7–11] Weekend surgery was associated with higher risks for death, postoperative respiratory and renal failure in both groups supported by the literature of mortality in colon[4] and general surgery.[5 6] Regarding

rectum resections, the literature reported insignificant effects. These are most likely explained by a small number of included cases.[4] These findings underline the need for the consideration of healthcare contexts in risk-adjusted quality assurance.

The hospital covariates in this analysis showed conflicting effects. Inconclusive results were found for rural localisation, university hospital status and hospital ownership. The estimated effects were insignificant (rural hospitals) or conflicting (volume, ownership) and therefore did not strongly affect the considered patient safety outcomes. The literature discusses the influence of case volume,[12 13 17 28 29] rural hospitals,[30–32] ownership,[29 33] university hospital status[34] or hospital size in general[29] with confirming or contradicting results often explained by, for example, patient case-mix, staffing or surgeon experience differing between hospital sizes.[29 35] This may be due to outcome-relevant information like staffing,[36 37] expertise[38] or certification[39] not being included in claims data. For example, a German study reported insignificant associations between ownership and postoperative wound infections after colon surgery.[40] The differences compared with our analyses are the procedure-definitions (partial/total colon resections vs open/laparoscopic colon procedure), the sample size (54 168 colon resections vs 28 291 colon procedures) and the data. The claims data used in our analysis include individual information on age, sex and comorbidities. Infection surveillance data used by Schröder et al does not include individual patient data on age, sex or severity of a patient's illness.[40]

Additionally, some studies did not stratify colon and rectum resections.[41 42] However, the heterogeneous results for both procedure groups indicate the relevance of stratification as already reported for other indications.[16]

With respect to case covariates age, sex and comorbidities like coagulopathies, heart diseases, lung diseases or fluid and electrolyte disorders were risk factors for poor patient outcome in both groups in this analysis, which is supported by the literature as well.[26 43–48]

There are several strengths to this study. This study analysed a large and current sample providing a broad span of cases, hospital types, ownerships and locations. While previous studies emphasised specific covariates and/or outcomes, we considered combined sets of previously solitarily analysed outcomes and risk factors and, thus, provide a comprehensive analysis. The applied multilevel-regression model is able to simultaneously analyse individual covariates like comorbidities and hospital-level covariates like annual case volume. It also considers relationships between covariates (eg, weekend surgery and emergency admissions).[25]

There are several limitations to this study. Secondary data induce challenges for a reliable operationalisation of outcomes. First, the data are anonymised. The anonymisation makes it impossible to validate the coded diagnoses.[49] Second, claims data do not include information which of the coded diagnoses had been present on admission. To overcome these shortcomings, this study used a

set of previously validated outcomes that were reported to occur most likely during hospitalisation.[23] With respect to transfer from other hospitals, recent literature distinguishes between urgent and non-urgent inter-hospital transfers.[9–11] The data included in this analysis does not include details on the reasons for transfer from other hospital. However, our results were adjusted for age, sex, comorbidities and weekend surgery representing severity and complexity. The different results depending on adjustment, stratification, bivariate and multivariate analyses underline the need for careful and comprehensive statistical analysis. One weakness of German hospital discharge data is a lack of information on patient history, medication, length of anaesthesia, staff-to-patient ratios, surgeon volumes, acuity/reasons for inter-hospital transfers, validity of coding, centralisation and comorbidities present on admission.[37 39 50] This lacking information may lead to bias as these covariates may influence the outcome and could not be considered in our study. To overcome these challenges we sought to define colon and rectum resections,[22] outcomes[23] and comorbidities[20] based on study literature for transparency and consistency. The advantage of this process has its limits. These definitions do not involve specific distinctions referring to procedure (eg, type, localisation) or comorbidities (eg, bowel disease). To ensure transparency we decided against creating our own definitions of procedures or comorbidities. An additional limitation is the limited possibility to analyse some specific subgroups (eg, case volume stratified by ownership, weekend surgeries stratified by admission) in models using a large set of covariates. It poses the risk of separation due alone to the small sample size of specific subgroup-populations and outcomes.[51]

## Conclusions

This study demonstrated that patient safety in colorectal resections is strongly related to specific healthcare covariates. Our results implicate a need to account for admission reasons and weekend surgery when measuring and comparing patient safety. Therefore a risk adjustment for these covariates in quality assurance measures should be pursued. Hospital volume, ownership, urbanisation degree and university hospital status could not be shown to be strongly associated with all patient safety outcomes of colorectal resections. Given these insights from an analysis of a large data set, this paper contributes reliable and comprehensive evidence to the ongoing debate on hospital- and healthcare-related influences on patient safety in general.

**Contributors** FW designed the concept, methods and investigation, visualised results and wrote the draft of the underlying analysis. MR undertook the formal analysis. MR and OS curated the data, supervised the methodology and project administration in general and revised the drafts. JS, ME-G, PS and RK acquired funding, supervised the concept and investigation and revised the drafts. FW is responsible for the overall content as the guarantor.

**Funding** The IMPRESS study is funded by the Innovation Fund of the Joint Federal Committee (Gemeinsamer Bundesausschuss, G-BA), Germany. Funding number: 01VSF16013. Open Access Funding by the Publication Fund of the TU Dresden.

**Competing interests** PS and RK are members of the scientific advisory board of IQM. ME-G serves as an external expert for IQM. The other authors declare that they have no conflict of interest.

**Patient and public involvement** Patients and/or the public were not involved in the design, or conduct, or reporting, or dissemination plans of this research.

**Patient consent for publication** Not applicable.

**Ethics approval** This study involves human participants and was approved by the ethics committee of the TU Dresden: IRB00001473 and IORG0001076.

**Provenance and peer review** Not commissioned; externally peer reviewed.

**Data availability statement** Data may be obtained from a third party and are not publicly available. The data are not publicly available due to containing information that could compromise research participant privacy/consent. The consent given from the research participants included the obligation to analyse the data anonymously without disclosure to other parties.

**ORCID iDs**
Felix Walther http://orcid.org/0000-0002-5259-124X
Olaf Schoffer http://orcid.org/0000-0001-6922-7148
Martin Roessler http://orcid.org/0000-0002-4662-4156

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
