## [Reviewer comments · BMJ Open]

ARTICLE DETAILS

TITLE (PROVISIONAL)	The relationships between multiple patient safety outcomes and healthcare and hospital-related risk factors in colorectal resection cases: Cross-sectional evidence from a nationwide sample of 232 German hospitals
AUTHORS	Walther, Felix; Schmitt, Jochen; Eberlein-Gonska, Maria; Kuhlen, Ralf; Scriba, Peter; Schoffer, Olaf; Roessler, Martin

VERSION 1 – REVIEW

REVIEWER	J Alverdy University of Chicago
REVIEW RETURNED	27-Nov-2021

GENERAL COMMENTS	Please succinctly state the limitations of the study, start like this There are several limitations to this study..... Please shorten the MS by at least 15%.
--

REVIEWER	Emilio De Raffele Azienda Ospedaliero-Universitaria di Bologna, Unità Operativa di Chirurgia Generale, Dipartimento dell'Apparato Digerente
REVIEW RETURNED	17-Dec-2021

GENERAL COMMENTS	The authors present a large observational study on a recent population of 54,168 colon resection cases from 209 German hospitals and 20,395 rectal resection cases from 200 German hospitals, to evaluate the impact of multiple covariates on patient clinical outcome, including level of care, hospital and case covariates. The manuscript is well thought out and well written. The conclusions on the risk factors and clinical outcome of colorectal surgery confirm most of the data from previous studies, which however examined more limited number of patients and specific covariates. The most relevant limitations of the study are related to the lack of significant information on patient history and comorbidities, centralization, surgeons involved in colorectal surgery in specific contexts (eg rural, university and private hospitals), intraoperative data. All of these covariates can lead to bias and ultimately affect the outcome. However these limitations are underlined and discussed by the authors.
--

REVIEWER	Sandra Nagl University Hospital Augsburg
REVIEW RETURNED	13-Mar-2022

GENERAL COMMENTS	Thank you for inviting me to review the study "Patient safety in
--

	colorectal resections – evidence from a nationwide sample”. This is an observational study analysing the relationship between multiple patient safety outcomes, such as in-hospital death, post-operative respiratory failure, renal failure and post-operative wound infections, with different care, hospital and case characteristics in a multilevel logistic regression model in a large sample of 54,168 colon resection cases and 20,395 rectum resection cases. The strength of this observational study lies in its large sample and the analysis of various outcome indicators and risk factors. However, important information is missing about the specific type of resection, the resection indications and some important medical history data and medication and the stratification of the analysed risk factors to these. Instead of performing a stratified analysis for cases with and without a cancer diagnosis, would it not be more useful to stratify the analysis by the precise diagnosis or indication for resection and the precise type of resection procedure? Another concern is how did you distinguish whether patients were referred from another hospital with or without an emergency indication? A transfer from another hospital is often due to emergency situations. A weekend operation is often based on an emergency. Did you have any overlap between these entities? How do you explain the lower risk of post-operative wound infections with non-profit or private ownership compared to treatment in university hospitals for colon resections? It seems unexpected that rectal resection did not show a significant association with non-profit or private ownership or university hospital and patient safety outcomes. Could you adapt the annual case volume to the hospital ownership/university hospital? Could you also summarize in the text, which comorbidities of the Elixhauser group were risk factors for poor patient outcomes. Intestinal bowel disease should also be analysed as an additional risk factor for patient outcomes.
--	---

VERSION 1 – AUTHOR RESPONSE

Reviewer: 1 - Dr. J Alverdy, University of Chicago, University of Chicago Biological Sciences Division

Comments to the Author:

Please succinctly state the limitations of the study, start like this There are several limitations to this study..... Please shorten the MS by at least 15%.

Thank you for your important advice. We reorganised the strengths and limitations section with the introducing phrases you mentioned. We also critically reviewed our manuscript for redundancies and removed 313 words (≈11%) initially. However, at the same time we need to mention that we added more passages due to the feedback of Reviewer 3. This particularly concerns the sections strengths and limitations and the case covariates.

Reviewer: 2 - Dr. Emilio De Raffe, Azienda Ospedaliero-Universitaria di Bologna

Comments to the Author:

- The authors present a large observational study on a recent population of 54,168 colon resection cases from 209 German hospitals and 20,395 rectal resection cases from 200 German hospitals, to evaluate the impact of multiple covariates on patient clinical outcome, including level of care, hospital and case covariates. The manuscript is well thought out and well written. The

conclusions on the risk factors and clinical outcome of colorectal surgery confirm most of the data from previous studies, which however examined more limited number of patients and specific covariates. The most relevant limitations of the study are related to the lack of significant information on patient history and comorbidities, centralization, surgeons involved in colorectal surgery in specific contexts (eg rural, university and private hospitals), intraoperative data. All of these covariates can lead to bias and ultimately affect the outcome. However these limitations are underlined and discussed by the authors.

Thank you for your positive feedback. Based on all reviewer and Editor comments we reorganised parts and content of our manuscript highlighted in tracked-changes mode.

Reviewer: 3 - Dr. Sandra Nagl, University Hospital Augsburg
Comments to the Author:

- Thank you for inviting me to review the study “Patient safety in colorectal resections – evidence from a nationwide sample”. This is an observational study analysing the relationship between multiple patient safety outcomes, such as in-hospital death, post-operative respiratory failure, renal failure and post-operative wound infections, with different care, hospital and case characteristics in a multilevel logistic regression model in a large sample of 54,168 colon resection cases and 20,395 rectum resection cases. The strength of this observational study lies in its large sample and the analysis of various outcome indicators and risk factors.
- However, important information is missing about the specific type of resection, the resection indications and some important medical history data and medication and the stratification of the analysed risk factors to these. Instead of performing a stratified analysis for cases with and without a cancer diagnosis, would it not be more useful to stratify the analysis by the precise diagnosis or indication for resection and the precise type of resection procedure?

Thank you for your query. We are aware of the advantages and especially the disadvantages of (German) accounting/ secondary data which we reported in our strengths and limitations section on the page 15. You mentioned the large sample size as a main advantage of secondary data. The advantage of a large data set on the other hand, inevitably requires a strictly defined and homogeneously structured set of variables. The consequence is a missing granularity.

Among other things, it is not known which diagnoses were present on admission, which indications/ diagnoses were directly related to surgical procedures, applied medication and a detailed medical history itself (pg. 15). Therefore, challenges appear in case of reliable definition of comorbidities, procedures and outcomes. In addition, medical care may be coded differently in German hospitals due to its accounting relevance.

To overcome these challenges we sought to referencably define colon and rectum resections, outcomes and comorbidities aiming at transparency and consistency. While ensuring validity and transparency, this approach also has some limitations. The applied definitions (German Inpatient Quality Indicators, Supplement S2-S3) distinguish between partial colon resections, total colon resections and rectum resections. They do not distinguish between more precise types of resection procedure as you mentioned above. To ensure validity of definitions, we decided not to define procedures on our own. Additionally, analyses of detailed subgroups pose the risk of insignificance or statistical separation due to small sample size. Nonetheless, we appreciate your query concerning this limitation and added this point to our limitations section (pg 15):

„[...] To overcome these challenges we sought to consistently and referencably define colon and rectum resections,22 outcomes23 and comorbidities20 for transparency. The advantage of this proceeding poses boundaries. These definitions do not involve specific distinctions referring to procedure (e.g. type, localization) or comorbidities (e.g. bowel disease). To keep transparency we decided to avoid own definitions of procedures or comorbidities. An additional limitation is the limited possibility to analyse detailed subgroups (e.g. case volume stratified by ownership, weekend surgeries stratified by admission) in models using a large set of covariates. It poses the risk of insignificance or separation due to smaller sample size of detailed subgroup-populations and outcomes.51 ...]“

- Another concern is how did you distinguish whether patients were referred from another hospital with or without an emergency indication? A transfer from another hospital is often due to emergency situations. A weekend operation is often based on an emergency. Did you have any overlap between these entities?

You are right. We are aware of the likely acuity inter-hospital transfers and weekend surgeries pose. Statistically, our multiple regression is able to handle these overlaps. In addition, weekend surgery may also occur when regular admissions become acutely ill. Therefore we did not stratify weekend surgeries into intransfer from other hospitals, emergency or referral admissions. As mentioned above, weekend surgeries (about 8 % in colon and 4% in rectum resections) pose the risks of statistical separation and insignificance due to its small numbers in our sample. In the additional phrase in our discussion we mentioned this example (pg. 15).

„[...] An additional limitation is the limited possibility to analyse detailed subgroups (e.g. case volume in private hospitals, emergency admissions in weekend surgeries) in models using a large set of covariates. It poses the risk of insignificance or separation due to smaller sample size of detailed subgroup-populations and outcomes.50...]“

- How do you explain the lower risk of post-operative wound infections with non-profit or private ownership compared to treatment in university hospitals for colon resections?

Thank you for this important query. We assume a misunderstanding. Therefore we would like to describe the results for a common understanding in more detail.

University hospital as risk factor: We compared 8 university hospitals included in our analysis against 224 non-university hospitals adjusting for all types of ownership. Referring to post-operative wound infections our analysis revealed a higher OR (1.981*, 95% CI 1.171 - 3.352) for post-operative wound infections in university hospitals compared to treatment in non-university hospitals. For better understanding we added the comparator into the description of our results on page 10:

„[...] Treatment in university hospitals was associated with increased odds of post-operative wound infections in colon (OR 1.98 (95% CI, 1.17-3.35)) and rectum resections (OR 2.29 (95% CI, 1.35-3.86)) compared to treatment in non-university hospitals....]“

Ownership as risk factor: We compared private and non-profit hospitals against public ownership adjusting for university hospitals. Compared to all public hospitals (including university and non-university hospitals), our analysis revealed a lower OR of post-operative wound infections in private (OR 0.777*, 95% CI 0.608 - 0.992) and non-profit (0.744* 95% CI 0.555 - 0.998) owned hospitals.

Thank you for reflecting the influence of ownership: A German study using national infection surveillance data described insignificant associations between hospital ownership and post-operative wound infections after colorectal surgery.1,2

The difference between our analyses are 1) the definitions of the analysed procedure (partial/ total colon resections vs. open/ laparoscopic colon procedure), 2) the sample size (54,168 colon resections

vs. 28,291 colon procedures) and 3) data including different information. The accounting data used in our analysis include individual data on age, sex and comorbidities while infection surveillance data used by Schröder et al. do not include individual patient data on age, sex or severity of a patient's illness.

1 Schröder C, Behnke M, Geffers C, et al. Hospital ownership: a risk factor for nosocomial infection rates? *Journal of Hospital Infection* 2018;100(1):76-82. doi: <https://doi.org/10.1016/j.jhin.2018.01.019>
2 Malheiro R, Peleteiro B, Correia S. Beyond the operating room: do hospital characteristics have an impact on surgical site infections after colorectal surgery? A systematic review. *Antimicrob Resist Infect Control* 2021;10(1):139. doi: 10.1186/s13756-021-01007-5 [published Online First: 2021/10/02]

We believe this is an important information. Therefore, we added this aspect to our discussion as it is an additional point able to explain differences between study results aiming at the same population and outcomes (pg. 14):

„[...]For example a German study reported insignificant associations between ownership and post-operative wound infections after colon surgery.⁴⁰ The differences between our analyses are the procedure-definitions (partial/ total colon resections vs. open/ laparoscopic colon procedure), the sample size (54,168 colon resections vs. 28,291 colon procedures) and the data. The accounting data used in our analysis include individual information on age, sex and comorbidities. Infection surveillance data used by Schröder et al. does not include individual patient data on age, sex or severity of a patient's illness.⁴⁰...]“

- It seems unexpected that rectal resection did not show a significant association with non-profit or private ownership or university hospital and patient safety outcomes. Could you adapt the annual case volume to the hospital ownership/ university hospital?

Thank you for your query. Unfortunately, we face the same problems mentioned above regarding small case numbers and risks of separation. Especially rectum resections include small numbers of critical outcomes like in-hospital death (n=870, 4.26%), post-operative respiratory failure (n= 2,494, 12.22%), renal failure (n=2,116, 10%) or post-operative wound infections (n=2,286, 11.20%). We explained this problem in our discussion section as mentioned above (pg.15).

„[...]An additional limitation is the limited possibility to analyse detailed subgroups (e.g. case volume stratified by ownership, weekend surgeries stratified by admission) in models using a large set of covariates. It poses the risk of insignificance or separation due to smaller sample size of detailed subgroup-populations and outcomes.⁵¹...]“

- Could you also summarize in the text, which comorbidities of the Elixhauser group were risk factors for poor patient outcomes.

We included the results of the Elixhauser comorbidities significantly associated with poor patient safety outcomes in descriptive and regression results and discussion.

Pg. 7: „[...]For Elixhauser comorbidities, the most frequent codes were solid tumor without metastasis (colon: 47.3%, rectum: 67.0%) uncomplicated hypertension (Colon/ rectum: 47%) and fluid and electrolyte disorders (colon: 45%, rectum: in both procedure groups. ...]“

Pg. 11: „[...] Of all Elixhauser comorbidities analysed, coagulopathies showed highest ORs for poor patient safety outcomes associated including higher ORs of death in colon (OR 4.17 (95% CI 3.864 - 4.509)) or rectum (OR 4.30 (95% CI 3.600 - 5.158)) resections for example. The same applies to other patient safety outcomes like post-operative respiratory failure in colon (OR 3.117 (95% CI 2.920 - 3.327)) or rectum (OR 3.052 (95% CI 2.697 - 3.455)) resections, renal failure in colon (OR 3.332 (95% CI 3.118 - 3.561)) and rectum (OR 2.886 (95% CI 2.541 - 3.277)) resections and post-operative

wound infections in colon (OR 1.644 (95% CI 1.531 - 1.764)) or rectum (OR 1.770 (95% CI 1.570 - 1.996)) resections either. Along with coagulopathies, fluid and electrolyte disorders, peripheral vascular disorders, congestive heart failure, chronic pulmonary disease, cardiac arrhythmias and pulmonary circulation disorders were also associated with multiple poor patient outcomes in both procedure groups. ...]"

Pg. 14: „[...]With respect to case covariates age, sex and comorbidities like coagulopathies, heart diseases, lung diseases or fluid and electrolyte disorders were risk factors for poor patient outcome...]"

- Intestinal bowel disease should also be analysed as an additional risk factor for patient outcomes.

Thank you for your query. For adjustment, we used the Elixhauser comorbidities. These do not include bowel diseases separately on the one hand. On the other hand, we are in need to use a predefined, validated and referenced comorbidity set (Elixhauser) applicable for secondary data for methodological issues. An individual addition or removal of ICD codes would not guarantee this consistency. The Elixhauser was specially developed for the analysis and adjustment of routine data analyses. We added this explanatory note as a limitation to our discussion section.:

„[...]we sought to consistently define colon and rectum resections,22 outcomes23 and comorbidities20 aiming at transparency. The advantage of this proceeding poses several boundaries. These definitions do not involve specific distinctions referring to procedure (e.g. type, localization) or comorbidities (e.g. bowel disease). To keep the transparency we decided not to define procedures or commodities on our own...]"

VERSION 2 – REVIEW

REVIEWER	Emilio De Raffe Azienda Ospedaliero-Universitaria di Bologna, Unità Operativa di Chirurgia Generale, Dipartimento dell'Apparato Digerente
REVIEW RETURNED	24-Jun-2022

GENERAL COMMENTS	The manuscript is well thought out and well written. The authors reasonably answered to the comments of the reviewers. As previously underlined, the most relevant limitations of the study are related to the lack of significant information on patient history and comorbidities, centralization, surgeons involved in colorectal surgery in specific contexts (eg rural, university and private hospitals), intraoperative data. All of these covariates can lead to bias and ultimately affect the outcome. However these limitations are related to the data sources and are underlined and discussed by the authors.
--

REVIEWER	Sandra Nagl University Hospital Augsburg
REVIEW RETURNED	13-Jun-2022

GENERAL COMMENTS	The authors have addressed and clarified the mentioned points well. The paper can be accepted in its current version.
--